# Microsporogenesis, Pollen Ornamentation, Viability of Stored *Taxodium distichum* var. *distichum* Pollen and Its Feasibility for Cross Breeding

Ziyang Wang [1,†], Ming Yin [2,†], David L. Creech [3] and Chaoguang Yu [1,*]

1 Jiangsu Engineering Research Center for Taxodium Rich, Germplasm Innovation and Propagation, Institute of Botany, Jiangsu Province and Chinese Academy of Sciences, Nanjing 210014, China; wangziyang@cnbg.net

2 Shanghai Academy of Landscape Architecture Science and Planning, Shanghai 200232, China; mingyin923@yahoo.com

3 Department of Agriculture, Arthur Temple College of Forestry and Agriculture, Stephen F. Austin State University, Nacogdoches, TX 75962-3000, USA; dcreech@sfasu.edu

* Correspondence: yuchaoguang888@sina.com; Tel.: +86-025-8434-7069

† These authors contributed equally to this work.

**Abstract:** *Taxodium* Rich is well known for its flooding tolerance and has great ecological and economic potential. A comprehensive understanding of pollen characteristics and storage capacity is important for breeding and genetic resource conservation of the genus. In this study, we observed the microsporogenesis and pollen ornamentation, studied the conditions of in vitro pollen germination, compared the difference in pollen viability of *T. distichum* var. *distichum* measured by in vitro germination and TTC staining, analyzed the change in pollen viability after different storage times and the feasibility of using stored pollen for cross breeding. Results indicated that the pollen mother cells of *T. distichum* var. *distichum* begin to enter the meiosis stage one month before the male strobilus disperse, reach metaphase 10 days after meiosis and form pollen grains three to five days after tetrad development. Pollen germination rate topped in the culture medium of 0.012% boric acid under 30 °C after 48 h, reaching 66.81%. The TTC staining demonstrated that the pollen viability of *T. distichum* var. *distichum* TD-4 and TD-5 were 97.78% and 80.54%, 98.96% and 91.67%, and 83.67% and 21.75% after one-, two- and three-year storage at −20 °C, which is significantly higher than ($p < 0.05$) that of 17.02 and 27.04%, 2.77% and 12.82%, and 0 determined by the in vitro cultivation. It is feasible to use pollen of *T. distichum* var. *distichum* TD-4 and TD-5 stored at −20 °C for one year for artificial hybridization, and the fruit setting rate and seed germination rate were 10.94 and 36.79%, and 11.47 and 65.76%, respectively.

**Keywords:** *Taxodium distichum* var. *distichum*; microsporogenesis; low temperature storage; pollen germination; hand pollination

## 1. Introduction

Pollen is the male gametophyte that carries plant genetic information and directly affects the breeding efficiency of crops [1,2], vegetables [3,4], fruits [5,6] and trees [7,8]. Its viability has an important impact on fertilization [9], embryo development [10] and seed quality [11]. The viability and longevity of plant pollen varies with species and environmental conditions [12]. The three main methods to test pollen viability include staining techniques [13–16], capacity to affect seed set [17,18] and in vivo and in vitro germination [19,20]. These methods each have advantages and disadvantages. The staining technique is simple and quick, but its results are often overestimated and may not show any correlation to the germination tests [21]. Determining pollen viability by seed set is the most authentic and accurate test, but it is environmentally sensitive, laborious and time consuming [12]. In vitro germination simulates natural pollination, so its detection results

correlate with the fruit and seed sets [12,22]. Pollen culture in vitro allows not only the germination rate of pollen to be observed, but also the development of the pollen tube to be monitored [12]. However, the conditions of pollen germination in vitro vary with different plant species, so it is necessary to study the optimal medium composition and concentration for each species [1,23,24]. Artificial culture media are dependent on species, and examples include soybean requires 15 g of sucrose, 0.03 g of $Ca(NO_3)_2$, and 0.01 g of $H_3BO_3$ in 100 mL of deionized water [1], the medium of *Vitis vinifera* which contains 15% sucrose and 1% agar [23], and the optimal medium for pollen germination of *Keteleeria fortunei* contains 240 g/L sucrose, 70 mg/L $CaCl_2$ and 210 mg/L $H_3BO_3$ [24].

The genus *Taxodium* represents deciduous or semi-evergreen trees of the Taxodiaceae, native to America, Mexico and Guatemala, and the genus is recognized as containing two main species: *T. distichum* var. *distichum*, *T. distichum* var. *imbricatum* and *T. mucronatum* [25,26]. As each species has known characteristics, *T. distichum* var. *distichum* and *T. distichum* var. *imbricatum* are used for their good form and excellent tolerance to waterlogging and needle blight, *T. mucronatum* has a long green leaf period and good tolerance to saline, alkali and drought [26,27] and *T.* 'zhongshanshan' is an interspecies hybrid clone generated from *Taxodium*, which combines the best characteristics of superior parents and shows outstanding heterosis. *T.* 'zhongshanshan' are versatile and used in river networks and wetlands, ornamentally in urban areas and for timber production in southeastern China [28]. The important characteristics of the genus, such as its long green leaf period, huge growth rate and the saline and alkali tolerance of *T. mucronatum*, are passed on in the matrilineal inheritance of the interspecies heterosis [27]. The hybrids of *T. mucronatum* × *T. distichum* var. *distichum* have great heterosis potential and artificial cross breeding is the key step in the selection and breeding of superior *T.* 'zhongshanshan' [27]. However, it is difficult to complete the pollination and fertilization of the *T. mucronatum* × *T. distichum* var. *distichum* cross because the flowering period of *T. distichum* var. *distichum* is up to one month later than that of *T. mucronatum*. As such, pollen must be stored for a year for cross pollination, and it is necessary to determine the viability of stored *T. distichum* var. *distichum* pollen and its feasibility for cross breeding.

Previous studies have focused on variety breeding, cutting propagation and the stress resistance physiology of *Taxodium* species [27,28], but only a few studies have considered its reproduction, such as the observation of pollen morphology [25] and embryo development [29,30]. Furthermore, the study of the flowering phenology, pollen development, pollen viability and cross compatibility of *Taxodium* species is limited. In this study, we observed the development of the male strobilus and pollen ornamentation, studied the conditions of in vitro pollen germination, compared the difference in pollen viability of *T. distichum* var. *distichum* measured by in vitro germination and TTC staining, analyzed the change in pollen viability after different storage time and the feasibility of using stored pollen for cross breeding, aiming to lay a foundation for reproductive biology research, germplasm resource preservation and cross breeding of *Taxodium* species.

## 2. Materials and Methods

### 2.1. Plant Materials

All plant materials came from five genotypes of *Taxodium* species, of which two genotypes (TD-4 and TD-5) from *T. distichum* var. *distichum* and three genotypes (TM-A2, TM-A5 and TM-B6) came from *T. mucronatum*. The five genotypes planted are listed in the National Tree Germplasm Bank of *Taxodium* 32°3′13.85″ N, 118°49′42.16″ E.

### 2.2. Observation of the Development of the Male Strobilus of T. distichum var. distichum

From 6 January to 13 March 2020, thirty male strobili of TD-5 were collected and measured daily. Then, they were fixed with a Carnot fixation solution consisting of $V_{95\%}$ alcohol combined with V glacial acetic acid in a three-to-one ratio and stored in a refrigerator at −20 °C [31]. The development process of microspores was observed through the method of conventional squashing followed by Giemsa-staining [32,33], and photographed under

an Olympus Vanox microscope at 100×. During microscopic examination, 30 pollen mother cells were randomly observed in each field, and at least 200 pollen mother cells were observed in each period.

On 16 March, the pollen grains of *T. distichum* var. *distichum* TD-5 was collected, and a series of treatments were then performed, including 4% glutaraldehyde fixation, cleaning with 0.1M phosphoric acid buffer at a pH of 7.2, gradient dehydration at 15 min intervals with ethanol at 30%, 50%, 70%, 90% and 100% and isoamyl acetate replacement and drying in HMDS (hexamethyldisilazane). Dried pollen grains were mounted on aluminum stubs and sputter coated with gold–palladium [5,25]. The surface patterns of the pollen samples were observed and photographed under a scanning electron microscopy (SEM) in a FEI QUANTA 200 microscope at 4000× and 10,000× with an acceleration voltage of 20 kV.

## 2.3. Determination of Pollen Water Content

The water content of fresh pollen of TD-4 and TD-5 collected on 18 March 2020 were determined in the same day. The water content of pollen collected in 2018, 2019 and 2020 and stored in a refrigerator at $-20\ °C$ were determined on 20 March 2021. Additionally, the water content of pollen was calculated according to Souza et al. [5]. Take a dry weighing bottle and mark the weight as A. Then, place 0.05 g of pollen in the weighing bottle, and denote the total weight as B. Then, open the cap of the weighing bottle and put it into the oven. After drying at 103 °C for 4 h, put the capped weighing bottle into the dryer for cooling, and weigh the total weight of pollen and weighing bottle. Weigh the dried pollen repeatedly after cooling until constant weight, denoted as C. Water content of pollen = (B − C/B − A) × 100%. Each treatment was repeated 3 times and the mean value was taken.

## 2.4. Pollen Germination Medium

On 18 March 2020, an orthogonal experiment of $L_{16}$ ($4^2$) was performed using a liquid culture of the fresh pollen of *T. distichum* var. *distichum* TD-5 collected in the same day with four sucrose concentrations of 0, 5%, 10% and 15% and four boric acid concentrations of 0, 0.005%, 0.01% and 0.015%. Based on the results, a single factor experiment of boric acid was carried out, with the concentration gradient of 0.006%, 0.008%, 0.01%, 0.012%, 0.014%, 0.016%, 0.018%, 0.02%, 0.022%, 0.024%, 0.026%, 0.028% and 0.03%. Pollen was added into the culture solution at a concentration of 0.01 g/mL, mixed with a toothpick and cultured in the incubator at 30 °C and three biological replicates were performed for each treatment. After 12, 24, 48, 60, 72 and 96 h, 100 μL pollen solution was observed for germination under an Olympus Vanox microscope at 10× and 20×. More than five fields each slide and more than 30 pollen grains within each field were recorded to determine germination rate and the length of pollen tube, allowing the optimal culture condition of in vitro pollen germination to be defined.

## 2.5. Viability of Stored T. distichum var. distichum Pollen

On 20 March 2021, the in vitro germination method and TTC staining were used to determine the pollen viability of *T. distichum* var. *distichum* TD-4 and TD-5 pollen collected for three consecutive years in 2018, 2019 and 2020 and stored in a refrigerator at $-20$ °C. Pollen was mixed with 0.5% TTC solution at a concentration of 0.01 g/mL, followed by 15 min incubation at 35 °C. Furthermore, in order to understand the reason for the difference between the pollen viability measured by TTC staining and in vitro germination, TTC staining was used to dye the pollen after in vitro cultivation.

## 2.6. Feasibility of Stored Pollen for Cross Breeding

In February 2020, six cross combinations were carried out with three *T. mucronatum* TM-A2, TM-A5 and TM-B6 genotypes serving as the maternal plant and pollen of *T. distichum* var. *distichum* TD-4 and TD-5 that had been collected in 2019 and stored for one year at $-20$ °C serving as the pollen donor. A total of 50 female strobili were pollinated for

each combination and repeated three times. In October 2020, the fruit-setting rate was determined at the cone maturation, then, the cones were harvested and left to dry under indoor conditions. On 25 January 2021, the seeds are removed from the cones and soaked in tap water, replace clean water once a week. On 25 March, 100 filled seeds from each cross combination were selected for germination in an outdoor field, repeated three times and the seed germination rate was analyzed on 20 April. Fruit setting rate = (number of cones harvested in October/number of pollinated female strobili in February) × 100%; seed germination rate = (number of germinated seed/planted seed number) × 100% [22].

### 2.7. Statistical Analysis

The statistical analysis was performed employing SPSS 19.0 (IBM Corporation, Somers, CT, USA). Results were expressed as means ± standard errors. All data were analyzed with one-way analysis of variance (ANOVA) followed by Duncan's test at $p < 0.05$.

## 3. Results

### 3.1. Development of the Male Strobilus

The pollen mother cells (Figure 1A) of the male strobilus of *T. distichum* var. *distichum* TD-5 entered meiosis (Figure 1B,C) 26 to 31 days before pollen dispersal, when the color of the male strobilus was emerald green with a length of 2.87 ± 0.21 mm and a diameter of 1.74 ± 0.06 mm. The cytokinesis during the meiosis of the pollen mother cell in *T. distichum* var. *distichum* belonged to the simultaneous type. The tetrad stage developed 9 to 11 days after the beginning of meiosis and the length and diameter of the male strobilus were 3.9 ± 0.13 mm and 2.08 ± 0.08 mm, respectively. Tetrads underwent development for three to five days to form pollen grains (Figure 1I) and the color of the male strobilus became yellow green with a length of 4.38 ± 0.25 mm and a diameter of 2.2 ± 0.05 mm. Pollen started to disperse 12 to 14 days later and male strobili turned lemon yellow at this stage, with a length of 5.52 ± 0.17 mm and diameter of 2.57 ± 0.13 mm. The tetrad formed during meiosis of the *T. distichum* var. *distichum* pollen mother cell was mainly tetrahedral tetrad (Figure 1E), but could be linear (Figure 1D), cross (Figure 1F), quadrilateral (Figure 1G) or T (Figure 1H). The relationship between the stages of microsporogenesis and observing features of the staminate strobilus of *T. distichum* var. *distichum* is shown in Table 1.

The equatorial view and polar view of pollen grains of *T. distichum* var. *distichum* TD-5 by Scanning electron microscopy are shown in Figure 2. The surface of pollen grains is densely covered with microverrucate ornamentation, on which many granular protrusions are observed.

### 3.2. Water Content of T. distichum var. distichum Pollen

The determination result of water content of pollen of *T. distichum* var. *distichum* TD-4 and TD-5 are shown in Table 2. With the extension of storage time, the pollen water content of TD-4 and TD-5 decreased gradually, and the water content of fresh pollen was the highest, and the water content of pollen after three years storage was the lowest. In addition, after the first year of storage, pollen water content decreased the most, from 24.46% and 24.38% of fresh pollen to 14.35% and 15.48% of pollen after storage for one year, and pollen water content decreased by 41.33% and 36.51%, respectively.

### 3.3. Pollen Germination Medium

The germination rate of *T. distichum* var. *distichum* pollen cultured in media with various sucrose and boric acid concentrations is shown in Table 3. The germination rate on the medium supplemented only with boric acid was 27.89 to 58.23%, significantly higher ($p < 0.05$) than that on medium containing no boric acid or only supplemented with sucrose at zero to 7.38%. The germination rate on media at various boric acid concentrations was substantially different ($p < 0.05$), so boric acid at an appropriate concentration showed a positive effect on pollen germination, whereas sucrose was inhibitory.

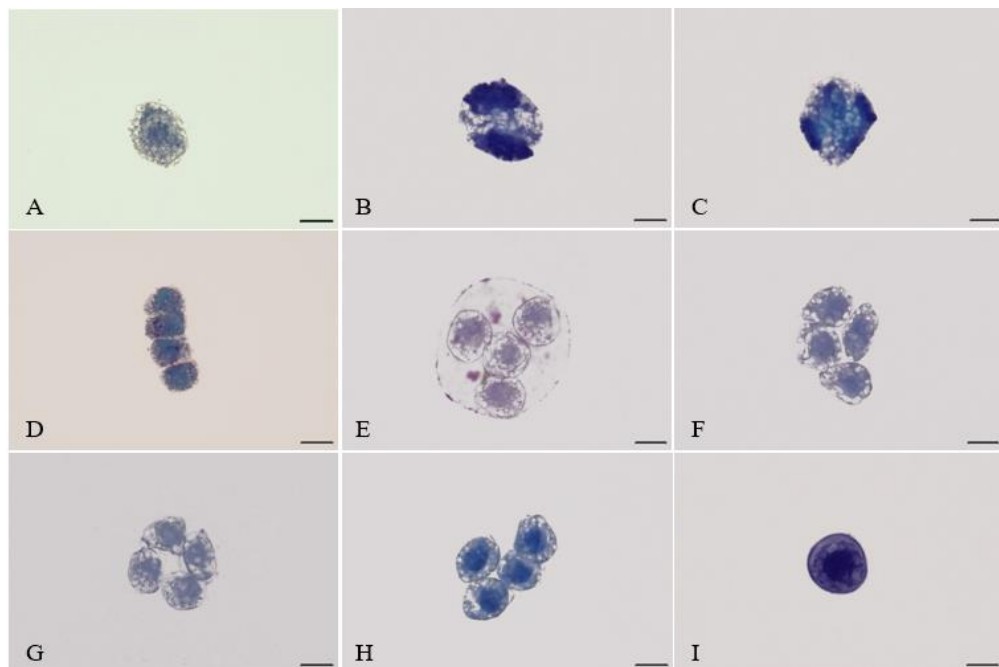

**Figure 1.** Stages of microsporogenesis in *T. distichum* var. *distichum*. (**A**) Microsporocyte. (**B**) Telophase I of meiosis. (**C**) Telophase II of meiosis. (**D**) Linear tetrad. (**E**) Tetrahedral tetrad. (**F**) Cross tetrad. (**G**) Quadrilateral tetrad. (**H**) The type T of tetrad. (**I**) Pollen grains. Bars: 10 μm.

**Table 1.** The relationship between the stages of microsporogenesis and observing features of the staminate strobilus of *T. distichum* var. *distichum*.

| Stages of Microsporogenesis | Color of | Single Strobilus | |
|---|---|---|---|
| | The Strobilus | Spike Length (mm) | Spike Diameter (mm) |
| Microsporocyte entered meiosis | Emerald green | $2.87 \pm 0.21$ | $1.74 \pm 0.06$ |
| Tetrad formation | May green | $3.9 \pm 0.13$ | $2.08 \pm 0.08$ |
| pollen grain formation | Yellow green | $4.38 \pm 0.25$ | $2.2 \pm 0.05$ |
| Pollen started to disperse | Lemon yellow | $5.52 \pm 0.17$ | $2.57 \pm 0.13$ |

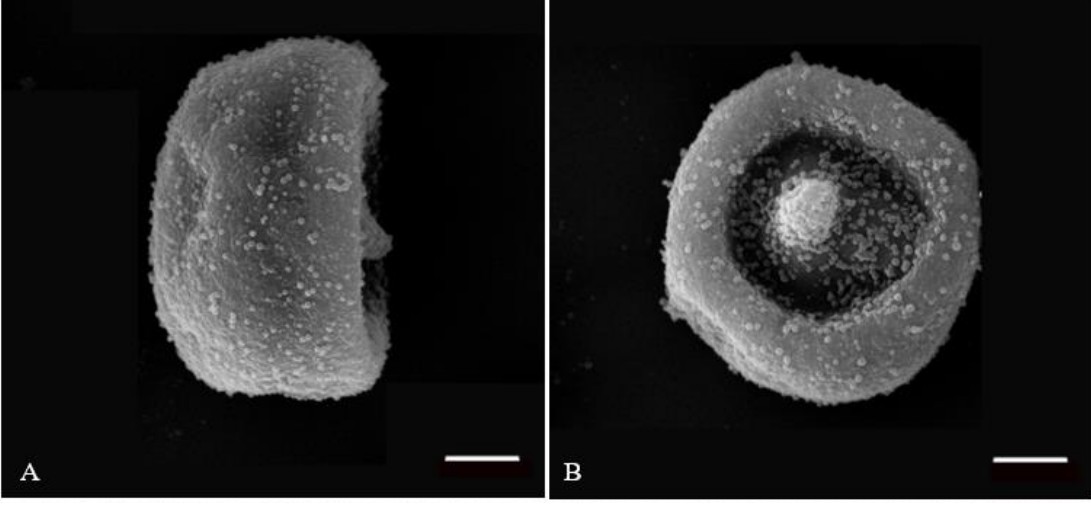

**Figure 2.** *Cont.*

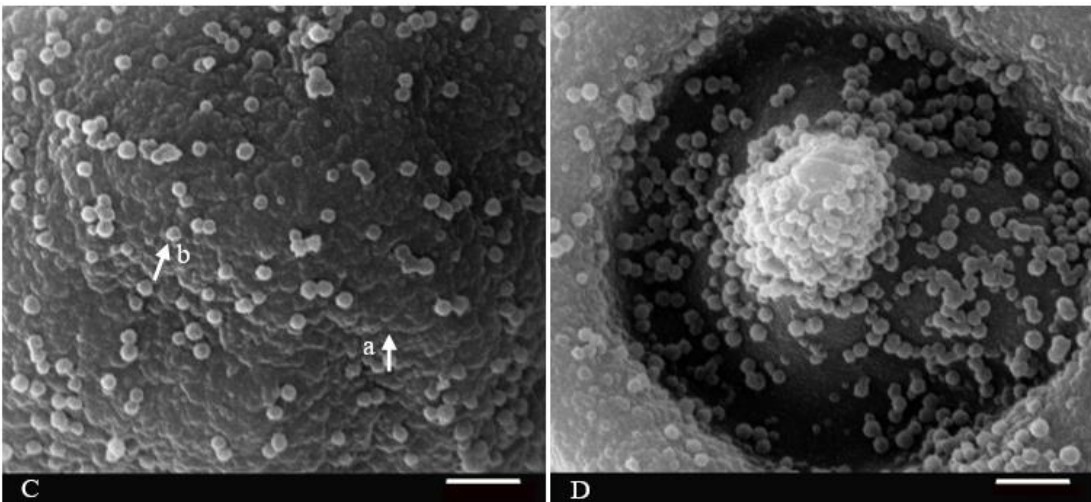

**Figure 2.** SEM photographs of pollen grains of *T. distichum* var. *distichum*. (**A**) Equatorial view at 4000×. (**B**) Polar view at 4000×. (**C**) Equatorial view at 10,000× showing pollen exine with microverrucate ornamentation (*arrow* a), on which many granular protrusions (*arrow* b) are observed. (**D**) Polar view at 10,000×. Bars: (**A,B**) 5 μm. (**C,D**) 2 μm.

**Table 2.** Variation of water content of *T. distichum* var. *distichum* pollen.

| Storage Time | *T. distichum* var. *distichum* TD-4 | *T. distichum* var. *distichum* TD-5 |
|---|---|---|
| | Water Content (%) | Water Content (%) |
| Fresh pollen | 24.46 ± 1.12 a | 24.38 ± 1.08 a |
| One-year storage at −20 °C | 14.35 ± 0.74 b | 15.48 ± 0.77 b |
| Two-year storage at −20 °C | 15.22 ± 0.83 b | 11.79 ± 0.81 c |
| Three-year storage at −20 °C | 8.90 ± 0.91 c | 12.42 ± 0.87 c |

Data are mean ± SE. Different lowercase letters in each column indicate significant differences between different storage time ($p < 0.05$).

**Table 3.** The effect of different mass fraction of sucrose and boric acid on pollen germination of *T. distichum* var. *distichum*.

| Treat | Sucrose Concentrations (%) | Boric Acid Concentrations (%) | Germination Rate (%) | Length of Pollen Tube (um) |
|---|---|---|---|---|
| 1 | 0 | 0 | 0.00 ± 0.00 e | 0.00 ± 0.00 d |
| 2 | 0 | 0.005 | 27.89 ± 6.90 c | 77.36 ± 11.09 b |
| 3 | 0 | 0.01 | 58.23 ± 4.67 a | 93.4 ± 18.27 a |
| 4 | 0 | 0.015 | 33.26 ± 6.59 b | 88.91 ± 21.50 a |
| 5 | 5 | 0 | 5.05 ± 1.99 d | 66.16 ± 11.10 c |
| 6 | 5 | 0.005 | 1.93 ± 0.89 de | 68.76 ± 8.50 c |
| 7 | 5 | 0.01 | 7.38 ± 4.21 d | 76.05 ± 11.49 b |
| 8 | 5 | 0.015 | 0.00 ± 0.00 e | 0.00 ± 0.00 d |
| 9 | 10 | 0 | 0.00 ± 0.00 e | 0.00 ± 0.00 d |
| 10 | 10 | 0.005 | 0.00 ± 0.00 e | 0.00 ± 0.00 d |
| 11 | 10 | 0.01 | 3.58 ± 2.44 de | 66.97 ± 10.70 c |
| 12 | 10 | 0.015 | 5.74 ± 1.71 d | 63.39 ± 6.14 c |
| 13 | 15 | 0 | 0.00 ± 0.00 e | 0.00 ± 0.00 d |
| 14 | 15 | 0.005 | 0.00 ± 0.00 e | 0.00 ± 0.00 d |
| 15 | 15 | 0.01 | 0.00 ± 0.00 e | 0.00 ± 0.00 d |
| 16 | 15 | 0.015 | 2.74 ± 0.79 de | 81.35 ± 7.79 ab |

Data are mean ± SE. Different lowercase letters in each column indicate significant differences between different mass fraction of sucrose and boric acid ($p < 0.05$).

The germination rate of *T. distichum* var. *distichum* pollen showed an increase followed by a decrease as the boric acid concentration increasing in the range of 0.006% to 0.03% (Table 4), ranging from 30.05% to 55.7% with a boric acid concentration of 0.006 to 0.01%. The highest pollen germination rate of 66.81% (Figure 3A) was observed on the media supplemented with 0.012% boric acid, which was significantly higher ($p < 0.05$) than that at other concentrations. At a boric acid concentration between 0.016 and 0.03%, the pollen germination rate was reduced from 55.3 to 16.64%. The range of pollen tube length was 82.04 to 93.65 μm and relatively long when the boric acid concentration was between 0.01% and 0.016%. At 0.006% and 0.028% boric acid, the length of the pollen tube was significantly shorter ($p < 0.05$) than at other boric acid concentrations. The data from this study indicated that pollen germination rate and length of pollen tube were significantly affected by boric acid concentration. The optimal boric acid concentration was 0.012%, with a pollen germination rate of 66.81%, average length of pollen tube of 90.52 μm and the longest pollen tube of 202.6 μm (Figure 3B). Extremely low and high concentrations of boric acid exhibited inhibitory effects on pollen germination and elongation of the pollen tube. During the in vitro cultivation of pollen, some pollen grains germinated two pollen tubes, part of which were branched from the same pollen tube (Figure 3C) and part of which were generated from two parts of the same pollen grain (Figure 3D).

**Table 4.** The effect of different mass fraction of boric acid on pollen germination of *T. distichum* var. *distichum*.

| Treat | Boric Acid Concentrations (%) | Germination Rate (%) | Length of Pollen Tube (um) |
|---|---|---|---|
| 1 | 0.006 | 30.05 ± 6.78 e | 82.04 ± 14.88 b |
| 2 | 0.008 | 34.50 ± 11.63 de | 87.04 ± 18.83 ab |
| 3 | 0.01 | 55.70 ± 9.99 b | 92.01 ± 11.23 a |
| 4 | 0.012 | 66.81 ± 12.16 a | 90.52 ± 18.18 ab |
| 5 | 0.014 | 47.81 ± 9.85 bcd | 93.65 ± 14.82 a |
| 6 | 0.016 | 55.30 ± 11.22 b | 93.51 ± 16.70 a |
| 7 | 0.018 | 51.62 ± 10.32 bc | 85.38 ± 24.00 b |
| 8 | 0.02 | 46.73 ± 12.37 bcd | 90.76 ± 13.29 ab |
| 9 | 0.022 | 43.59 ± 13.80 cd | 85.86 ± 17.37 ab |
| 10 | 0.024 | 38.46 ± 13.45 de | 89.67 ± 17.48 ab |
| 11 | 0.026 | 35.23 ± 11.57 de | 85.76 ± 15.34 ab |
| 12 | 0.028 | 32.43 ± 13.07 e | 82.55 ± 17.57 b |
| 13 | 0.03 | 16.64 ± 8.52 f | 90.27 ± 13.93 ab |

Data are mean ± SE. Different lowercase letters in each column indicate significant differences between different mass fraction of boric acid ($p < 0.05$).

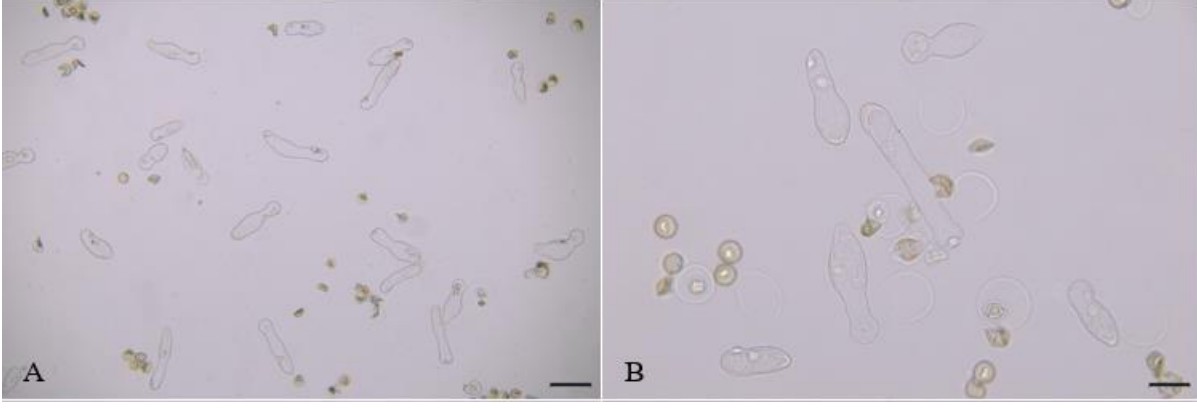

**Figure 3.** *Cont.*

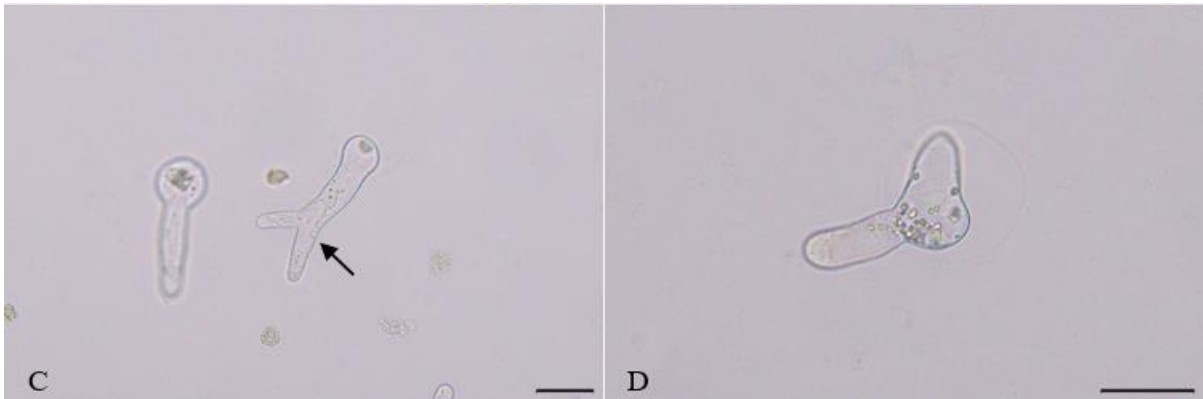

**Figure 3.** Pollen germination of *T. distichum* var. *distichum* in vitro. (**A**) Observation of pollen germination at 10×. (**B**) Observation of pollen germination at 20×. (**C**) Two tubes branched from the same pollen tube (*arrow*). (**D**) Two tubes generated from two parts of the pollen grain. Bars: (**A**) 100 µm. (**B–D**) 50 µm.

The variation of pollen germination of *T. distichum* var. *distichum* with a change in culture time is seen in Table 5. Pollen started to germinate 24 h after cultivation and no significant difference ($p < 0.05$) was shown for the germination rate and pollen tube length at 48, 60, 72 and 96 h, indicating that the optimal time for pollen in vitro cultivation was 48 h.

**Table 5.** Variation of pollen germination of *T. distichum* var. *distichum* with a change in culture time.

| Germination Time (h) | Germination Rate (%) | Length of Pollen Tubes (um) |
|:---:|:---:|:---:|
| 12 | 0.00 ± 0.00 d | 0.00 ± 0.00 d |
| 24 | 14.05 ± 1.69 c | 36.01 ± 6.12 c |
| 36 | 36.80 ± 3.55 b | 62.07 ± 5.11 b |
| 48 | 62.35 ± 11.61 a | 90.92 ± 11.91 a |
| 60 | 62.21 ± 8.83 a | 90.14 ± 11.52 a |
| 72 | 61.96 ± 9.84 a | 90.94 ± 17.12 a |
| 96 | 63.19 ± 16.22 a | 93.52 ± 14.65 a |

Data are mean ± SE. Different lowercase letters in each column indicate significant differences between different germination time ($p < 0.05$).

### 3.4. Viability of Stored Pollen Grains

Pollen of *T. distichum* var. *distichum* TD-4 and TD-5 did not germinate after being stored at −20 °C for three years and the germination rate of pollen after two-year storage were significantly smaller ($p < 0.05$) than those of one-year storage (Table 6). The germination rate of TD-4 was substantially less ($p < 0.05$) than that of TD-5 regardless of collecting time. The results indicated that pollen germination rate varied among *T. distichum* var. *distichum* with different genetic backgrounds and that *T. distichum* var. *distichum* pollen retained some germination potential after two-year storage at −20 °C and the appropriate storage duration was less than three years.

**Table 6.** Comparison of viability of pollen after storage at −20 °C of *T. distichum* var. *distichum* measured by TTC staining and in vitro germination.

| Determine Method | *T. distichum* **var.** *distichum* **TD-4** | | | *T. distichum* **var.** *distichum* **TD-5** | | |
|:---:|:---:|:---:|:---:|:---:|:---:|:---:|
| | One-Year | Two-Year | Three-Year | One-Year | Two-Year | Three-Year |
| TTC staining (%) | 97.78 ± 1.25 a | 98.96 ± 1.46 a | 83.67 ± 4.27 a | 80.54 ± 8.05 a | 91.67 ± 2.70 a | 21.75 ± 2.03 a |
| In vitro germination (%) | 17.02 ± 3.19 b | 2.76 ± 1.32 b | 0.00 ± 0.00 b | 27.04 ± 6.20 b | 12.82 ± 5.38 b | 0.00 ± 0.00 b |

Data are mean ± SE. Different lowercase letters in each column indicate significant differences between the two determine methods of TTC staining and in vitro germination ($p < 0.05$).

The staining of pollen of *T. distichum* var. *distichum* TD-5 after one-, two- and three-year storage and pollen after in vitro cultivation showed some pollen was stained dark red, some light red and some were not stained (Figure 4A,B). All germinated pollen from in vitro cultivation were stained red, but some red pollen did not germinate normally (Figure 4B). The TTC staining demonstrated that the pollen viability of TD-4 and TD-5 was 83.67% and 21.75% after three-year storage at −20 °C and all were over 80% after one to two years of storage, which is significantly higher than ($p < 0.05$) that determined by the in vitro cultivation method (Table 6).

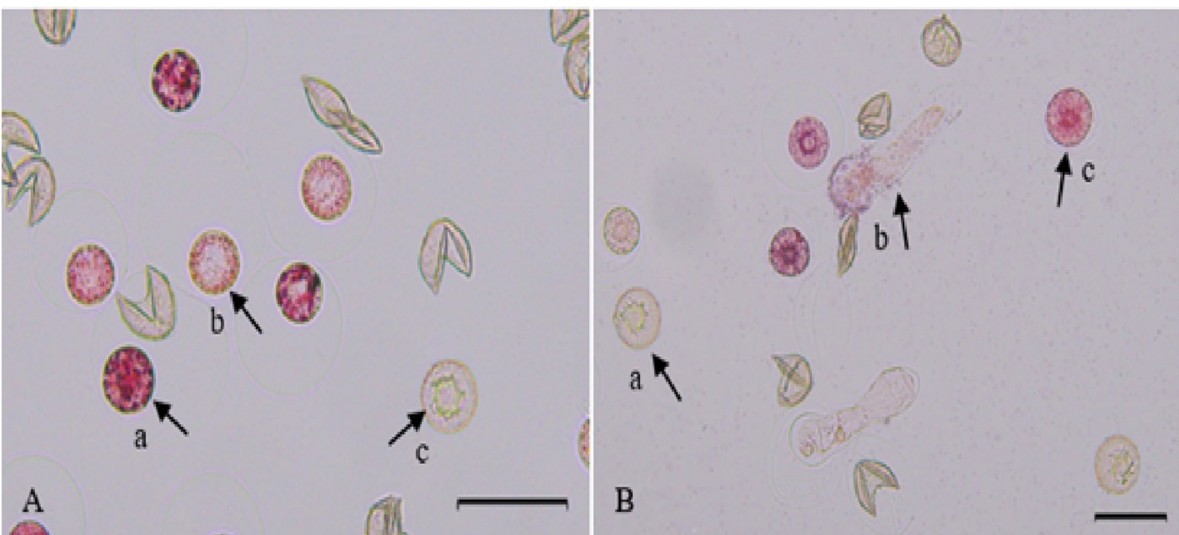

**Figure 4.** Pollen and pollen tubes of *T. distichum* var. *distichum* in TTC staining. (**A**) The staining of pollen grains showing pollen with strong viability (*arrow* a), pollen with moderate viability (*arrow* b) and the nonviable pollen (*arrow* c). (**B**) The staining of pollen grains after in vitro cultivation showing that the nonviable pollen were not stained (*arrow* a), the germinated pollen were stained red (*arrow* b) and the stained pollen did not germinate (*arrow* c). Bars: 50 μm.

*3.5. Feasibility of Stored Pollen for Cross Breeding*

The resulting fruit setting rate and seed germination rate using three *T. mucronatum* genotypes, TM-A2, TM-A5 and TM-B6 as maternal plants and one-year-old pollen of *T. distichum* var. *distichum* TD-4 and TD-5 stored at −20 °C as the pollen donor for hybridization, can be seen in Table 7. The fruit setting rate and seed germination rate with TD-4 pollen as donor were 28.13 to 36.79% and 11.47 to 26.79%, respectively, and 10.94 to 33.37% and 56.12 to 65.76% with TD-5 pollen, indicating that it was feasible to use pollen of *T. distichum* var. *distichum* for cross breeding after storing at −20 °C for one year.

**Table 7.** Statistics of fruit setting rate and seed germination rate using one-year-old pollen of *T. distichum* var. *distichum* TD-4 and TD-5 stored at −20 °C as the pollen donor for hybridization.

| Hybrid Combination | Fruit Set (%) | Seed Germination Rate (%) |
|---|---|---|
| TM-A2 × TD-4 | 36.79 ± 3.08 a | 24.67 ± 3.46 c |
| TM-A5 × TD-4 | 34.18 ± 2.32 a | 11.47 ± 2.29 d |
| TM-B6 × TD-4 | 28.13 ± 1.52 b | 26.79 ± 4.58 c |
| TM-A2 × TD-5 | 33.37 ± 1.97 a | 65.76 ± 3.80 a |
| TM-A5 × TD-5 | 10.94 ± 1.48 d | 61.32 ± 5.49 ab |
| TM-B6 × TD-5 | 20.65 ± 2.21 c | 56.12 ± 3.73 b |

Data are mean ± SE. Different lowercase letters in each column indicate significant differences between different hybrid combination ($p < 0.05$).

## 4. Discussion

The size and color of the male strobilus in *T. distichum* var. *distichum* TD-5 are closely related to the process of meiosis and it gradually develops and matures with an increase in microstrobilus and a color transition from emerald green to lemon yellow. The pollen mother cells of *T. distichum* var. *distichum* begin to enter the meiosis stage one month before the male strobilus disperse, reach metaphase nine to eleven days after meiosis and form pollen grains three to five days after tetrad development. These stages are synchronized in the same microstrobilus, whereas there are discrepancies in the development process of pollen mother cells in different inflorescences and microstrobilus, which is like the microsporogenesis of Taxodiaceae, such as Cryptomeria fortune [31], Cunninghamia lanceolata [32] and Metasequoia glyptostroboides [33]. In the present study, multiple types of tetrads formed during the meiosis of T. distichum var. distichum with tetrahedral as the major type, which might be due to the cytokinesis of meiosis belonging to the simultaneous type. Two daughter nuclei directly enter the second meiosis without cytokinesis in the first meiosis of microspore mother cell and cytokinesis is produced after the end of the second meiosis, consistent with the tetrad formation process in Picea koraiensis [34], Cryptomeria fortunei [31] and Cunninghamia lanceolata [32]. Observations by SEM showed that the surface of pollen grains of T. distichum var. distichum TD-5 is densely covered with granular protrusions, which is similar to the study by Tiwari et al. [25].

In vitro pollen germination is widely used because of its accuracy, simplicity and feasibility [12] and appropriate culture conditions are the key to pollen germination [5,35]. The requirement of nutrients and mineral elements supplemented in the culture medium varies for different plants and sucrose and boric acid are the two most common components in the pollen culture medium [24,36,37]. Sucrose is not only the main nutrient for pollen germination and pollen tube elongation, but also plays a role in maintaining the osmotic balance between pollen and culture medium to avoid the rupture of pollen and pollen tube [38]. Boric acid is rich in plant stigmas and ovaries, and it can increase the absorption, transportation and metabolism of pollen, promote biosynthesis of pectin for pollen tube membranes and enable the entry of extracellular calcium ions into cells to facilitate the transportation of nutrients [39]. In this study, the germination rate of *T. distichum* pollen on the medium with sucrose but without boric acid was less than 7.38%, but it was significantly increased ($p < 0.05$) at 27.89% to 58.23% on media supplemented with boric acid but without sucrose, indicating the promoting and inhibiting effect of boric acid and sucrose on the in vitro germination of *T. distichum* var. *distichum* pollen. With an increase in boric acid concentration in the medium, an increase in pollen germination rate followed by a decrease was observed, indicating that the appropriate concentration of boric acid can promote pollen germination, but it will inhibit the germination at a high concentration. Such a phenomenon is similar to the study on *Cunninghamia lanceolata* [40], *Keteleeria fortune* [24] and *Picea meyeri* [39].

The common use of TTC staining to test pollen viability is due to simplicity and speed [12]. The active pollen is quickly stained red because it contains peroxidase and the darker the color, the stronger the pollen viability and nonviable pollen cannot be stained [41]. Its accuracy is controversial, because although many studies support TTC staining to evaluate pollen viability [35,42–44], others found that pollen viability was overestimated due to the false positives in this test [37,45,46]. In the present study, pollen viability of *T. distichum* determined by TTC staining was significantly higher ($p < 0.05$) than that measured by the in vitro germination method. A large amount of pollen stained by TTC could not germinate normally, suggesting that it cannot accurately reflect the actual germination ability of pollen in *T. distichum* var. *distichum*, and the combination of TTC staining and in vitro germination is a more appropriate test for pollen viability in the breeding of *Taxodium* species.

Pollen storage capacity is related to its own genetic characteristics and external factors, and different pollen exhibit different longevity with various storage conditions, as well as different viability after certain periods of storage [5,22,47]. External factors, such as

temperature, humidity, light and storage medium all have an impact on the storage capacity of pollen, with temperature as one of the main factors affecting pollen viability [1,22]. Due to the decreased respiration of pollen, the consumption of soluble sugar and organic acid is reduced at low temperatures, so low temperature is generally used for the long-term preservation of pollen [48,49]. The study of pollen storage capability in two poplar species by Du et al. [22] found that the pollen germination rate was 23.77% and 34.19% after one year of storage at −20 °C, which reduced to 11.97% and 11.03% after two-year storage, but the seeds derived from crosses using pollen stored for two years developed normally. Fernando et al. [47] showed that the germination rate of *Castanea dentata* pollen was 8% and 19% after one-year storage at −20 °C and −80 °C, respectively. Souza et al. [5] found that the fruit setting rate of *Aechmea bicolor* after pollinating with pollen stored at −80 °C and −196 °C for one year was greater than 58%, from which normally germinated seeds were obtained. In the present study, the pollen germination rate of *T. distichum* var. *distichum* pollen stored for one and two years at −20 °C was 17.02 to 27.04% and 2.77% to 12.82%, respectively, and the fruit setting and seed germination rate was 10.94 to 36.79% and 11.47 to 65.76% after using one-year-old pollen stored at −20 °C for pollination. In conclusion, it is feasible to use *T. distichum* var. *distichum* pollen stored at −20 °C for one year for artificial hybridization, which can produce hybrid seeds with normal germination.

**Author Contributions:** Z.W. and M.Y. performed the experiments, analyzed the data and wrote the manuscript. D.L.C. participated in statistical analysis and manuscript preparation. C.Y. conceived the idea, proposed, initiated and led the project, interpreted scientific information and participated in manuscript preparation. All authors have read and agreed to the published version of the manuscript.

**Funding:** This study was supported by the Jiangsu Long-term Scientific Research Base for *Taxodium* Rich. Breeding and Cultivation [LYKJ(2021)05], and the Jiangsu Province Policy Guidance Program Special Project for Introducing Foreign Talents (BX2020010).

**Institutional Review Board Statement:** Not applicable.

**Informed Consent Statement:** Not applicable.

**Data Availability Statement:** Not applicable.

**Conflicts of Interest:** The authors declare no conflict of interest.

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
