# Peer review of "Microsporogenesis, Pollen Ornamentation, Viability of Stored Taxodium distichum var. distichum Pollen and Its Feasibility for Cross Breeding"

_forests, doi:10.3390/f13050694_

Round 1

Reviewer 1 Report

The paper adds new and interesting data to the current knowledge on the morphology and vitality of the stored pollen. The title is appropriate and informative. The introduction summarizes current knowledge and identifies gaps; also, the reason for conducting this research is clearly formulated. The methods are appropriate, but some of the analysis should be further clarified, please see the comments below. The results are comprehensively reported and mostly supported by figures and tables which are helpful for the understanding of the findings. The Discussion section should be supplemented with appropriate references.

Still, some issues need to be addressed:

Comments

  1. Line 52: reference
  2. Line 54: reference
  3. Line 97: Is it a good year mentioned in this line? The later time period does not match…
  4. Line 98: Only the genotype TD-5 is listed. When the TD-4 genotype was sampled, was the same methodology applied?
  5. Line 100: is the protocol known?
  6. Line 105: same question, the explanation for the TD-4 (above)
  7. Line 111: protocol (reference)
  8. Line 129: has only the pollen of the genotype TD-5 been tested?
  9. Line 141: it is not clear, pollen from which year was analyzed?
  10. Line154: specify the method of testing seed germination.
  11. Line 161: what is taken as a source of variability? It would be very useful to clarify the results obtained if a multifactorial analysis of variance was performed, where the influence of origin (genotype) and year of flowering of the investigated pollen traits would be determined.
  12. Line 174: it would be very useful for readers to give a tabular presentation of the stated morphological values of strobiles.
  13. Line 306-313: can this be claimed if the results are obtained only on the basis of two genotypes?The authors should determine the results based on the material used for the research.
  14. Line 331: can this be claimed if no research has been done related to the influence of origin (genotype)?
  15. Line 334: reference
  16. Line 350: reference
  17. Line 358: reference
  18. Line 372: reference

Author Response

Point 1: Line 52: reference

Response 1: The sentence mentioned by reviewer was supplemented with appropriate reference.

Point 2: Line 54: reference

Response 2: The sentence mentioned by reviewer was supplemented with appropriate reference.

Point 3: Line 97: Is it a good year mentioned in this line? The later time period does not match…

Response 3: The observation period was from January 6 to March 13 2020, and the corresponding modifications have been made in the revised version.

Point 4: Line 98: Only the genotype TD-5 is listed. When the TD-4 genotype was sampled, was the same methodology applied?

Response 4: The development of the male strobilus of other genotypes, such as TD-4, is similar to that of TD-5. And the observation method was the same for the other genotypes.

Point 5: Line 100: is the protocol known?

Response 5: The method mentioned by reviewer was supplemented with appropriate reference.

Point 6: Line 105: same question, the explanation for the TD-4 (above)

Response 6: The pollen morphology and observation method of other genotypes, such as TD-4, are the same as those of TD-5. And the part of pollen morphology was removed in the revised version according to the comments of reviewer.

Point 7: Line 111: protocol (reference)

Response 7: The part of pollen morphology was removed in the revised version according to the comments of reviewer.

Point 8: Line 129: has only the pollen of the genotype TD-5 been tested?

Response 8: Only the fresh pollen of T. distichum var. distichum TD-5 collected in 2020 has been tested for the screening of medium of in vitro pollen germination. Then, based on this test, the in vitro germination method was used to determine the pollen viability of TD-4 and TD-5 pollen after one-, two- and three-year storage at -20°C.

Point 9: Line 141: it is not clear, pollen from which year was analyzed?

Response 9: On March 20th, 2021, the in vitro germination method and TTC staining were used to determine the pollen viability of T. distichum var. distichum TD-4 and TD-5 pollen collected for three consecutive years in 2018, 2019 and 2020 and stored in a refrigerator at -20°C. And the corresponding modifications have been made in the revised version.

Point 10: Line154: specify the method of testing seed germination.

Response 10: The corresponding modifications have been made in the revised version.

Point 11: Line 161: what is taken as a source of variability? It would be very useful to clarify the results obtained if a multifactorial analysis of variance was performed, where the influence of origin (genotype) and year of flowering of the investigated pollen traits would be determined.

Response 11: All data in this study were analyzed with one-way analysis of variance (ANOVA) followed by Duncan’s test at P < 0.05, and multivariate analysis was not involved.  
Different lowercase letters in each column of table 1 indicated that the size of pollen grains of T. distichum var. distichum TD-5 had significant difference in different years (P<0.05), and the table 1 was removed in the revised version according to the comments of reviewer. 
Different lowercase letters in each column of table 2 indicate significant differences between different storage time (P<0.05). 
Different lowercase letters in each column of table 3 indicate significant differences between different mass fraction of sucrose and boric acid (P <0.05). 
Different lowercase letters in each column of table 4 indicate significant differences between different mass fraction of boric acid (P <0.05). 
Different lowercase letters in each column of table 5 indicate significant differences between different germination time (P <0.05). 
Different lowercase letters in each column of table 6 indicate significant differences between the two determine methods of TTC staining and in vitro germination (P <0.05).  
Different lowercase letters in each column of table 7 indicate significant differences between different hybrid combination (P <0.05). 
And The corresponding modifications have been made in the revised version.

Point 12: Line 174: it would be very useful for readers to give a tabular presentation of the stated morphological values of strobiles.

Response 12: The tabular presentation mentioned by reviewer was supplemented in the revised version.

Point 13: Line 306-313: can this be claimed if the results are obtained only on the basis of two genotypes?The authors should determine the results based on the material used for the research.

Response 13: The corresponding modifications have been made in the revised version.

Point 14: Line 331: can this be claimed if no research has been done related to the influence of origin (genotype)?

Response 14: The paragraph was removed in the revised version according to the comments of reviewer.

Point 15: Line 334: reference

Response 15: The sentence mentioned by reviewer was supplemented with appropriate reference.

Point 16: Line 350: reference

Response 16: The sentence mentioned by the reviewer is the author's hypothesis, which has been removed in the revised version.

Point 17: Line 358: reference

Response 17: The sentence mentioned by reviewer was supplemented with appropriate reference.

Point 18: Line 372: reference

Response 18: The sentence mentioned by reviewer was supplemented with appropriate reference.

Reviewer 2 Report

The manuscript deals with an interesting subject about the microsporogenesis, viability and morphology of the pollen grain of two Taxodium species. The Introduction is well written and shows the objectives of the paper. The Methodology concerning the part of microsporogenesis and viability is well detailed, however, the part of pollen morphology is missing and shows that the authors do not know the basic palynology.. The results focused on microsporogenesis and viability are clear and comprehensive, on pollen morphology, it is clear the lack of knowledge about the principles of Palynology established by Erdtman (1952), the terminology used, etc. See comments in the attached pdf. The same applies to the Discussion item. The figures are very good, particularly the pollen grain plate in scanning electron microscopy.
Based on the above, I believe that the manuscript could be accepted if the authors removed the pollen morphology, since, in order to present the dimensions of the pollen grains, a whole laboratory treatment would be necessary.

Author Response

Point 1: Line 2: The abbreviation of the Taxodium was marked in yellow.

Response 1: "T." was replaced with "Taxodium" in the revised version.

Point 2: Line 17: Review the sentence.

Response 2: The sentence mentioned by reviewer was rewrote in the revised version.

Point 3: Line 34: It is not interesting to repeat the title in the keywords.

Response 3: The corresponding modifications have been made in the revised version. 

Point 4: Line 83: Review the sentence.

Response 4: The sentence mentioned by reviewer was rewrote in the revised version.

Point 5: Line 115: The word microsporophyll was marked with “???”

Response 5: The corresponding modifications have been made in the revised version.  

Point 6: Line 115:The word papilla was marked with “???”

Response 6: The part of pollen morphology was removed in the revised version according to the comments of reviewer.  

Point 7: Line 181: To define the shape, size of the pollen grain as well as the dimensions of the aperture, it is necessary that the material has been subjected to the acetolysis process of Erdtman (1952, 1960). Under scanning electron microscopy we must not establish the diameters and, consequently, neither the shape nor the size defined by Erdtman.

Response 7: The part of pollen morphology was removed in the revised version according to the comments of reviewer.

Point 8: Line 322: The authors repeat the information already presented in the Results. There is no way to compare current results with those of past authors unless they also did not use acetolysed material.

Response 8: The part of pollen morphology was removed in the revised version according to the comments of reviewer.
